# Integration of a Physiologically Based Pharmacokinetic and Pharmacodynamic Model for Tegoprazan and Its Metabolite: Application for Predicting Food Effect and Intragastric pH Alterations

**DOI:** 10.3390/pharmaceutics14061298

**Published:** 2022-06-18

**Authors:** Hyeon-Cheol Jeong, Min-Gul Kim, Zhuodu Wei, Kyeong-Ryoon Lee, Jaehyeok Lee, Im-Sook Song, Kwang-Hee Shin

**Affiliations:** 1Research Institute of Pharmaceutical Sciences, College of Pharmacy, Kyungpook National University, Daegu 41566, Korea; houkiboshi01@knu.ac.kr (H.-C.J.); 2021221434@knu.ac.kr (Z.W.); here0723@gmail.com (J.L.); isssong@knu.ac.kr (I.-S.S.); 2Department of Pharmacology, Medical School, Jeonbuk National University, Jeonju 54907, Korea; mgkim@jbcp.kr; 3Laboratory Animal Resource Center, Korea Research Institute of Bioscience and Biotechnology, Cheongju 28116, Korea; kyeongrlee@kribb.re.kr; 4Department of Bioscience, University of Science and Technology, Daejeon 34113, Korea

**Keywords:** tegoprazan, PBPK, pharmacodynamics, gastric pH, food effect

## Abstract

A physiologically based pharmacokinetic/pharmacodynamic (PBPK/PD) model for tegoprazan and its major metabolite M1 was developed to predict PK and PD profiles under various scenarios. The PBPK model for tegoprazan and M1 was developed and predicted using the SimCYP^®^ simulator and verified using clinical study data obtained after a single administration of tegoprazan. The established PBPK/PD model was used to predict PK profiles after repeated administrations of tegoprazan, postprandial PK profiles, and intragastric pH changes. The predicted tegoprazan and M1 concentration–time profiles fit the observed profiles well. The arithmetic mean ratios (95% confidence intervals) of the predicted to observed values for the area under the curve (AUC_0–24 h_), maximum plasma drug concentration (C_max_), and clearance (CL) for tegoprazan and M1 were within a 30% interval. Delayed time of maximum concentration (T_max_) and decreased C_max_ were predicted in the postprandial PK profiles compared with the fasted state. This PBPK/PD model may be used to predict PK profiles after repeated tegoprazan administrations and to predict differences in physiological factors in the gastrointestinal tract or changes in gastric acid pH after tegoprazan administration.

## 1. Introduction

Proton-pump inhibitors (PPIs) are widely used for treating gastroesophageal reflux disease (GERD) and peptic ulcers or for eradicating *Helicobacter pylori* [1]. GERD is caused by the regurgitation of stomach contents into the esophagus, resulting from relaxation of the lower esophageal sphincter or an increase in gastric pressure [2]. PPIs, including omeprazole, pantoprazole, and esomeprazole, are primarily used to prevent damage to the esophageal epithelial tissues, caused by gastric acid reflux, in patients with GERD, Barrett’s esophagus, and peptic ulcer disease [3]. The plasma half-life of most PPIs is less than 2 h; however, the half-life of tenatoprazole is up to 9 h [1]. Although PPIs are widely used to treat gastric acid-related disorders, they have several limitations. PPIs, after their conversion to an active form in a low-pH environment in the parietal cell canaliculus, can inhibit H^+^/K^+^-ATPase; however, it takes 2–3 days to reach the steady-state of acid inhibition [1,4]. In addition, most PPIs exhibit large inter-individual variability and potential for high drug–drug or drug–genetic interactions because they are primarily metabolized by CYP2C19 [5]. The frequency of CYP2C19 poor metabolizers (PM) is approximately 2–6% in Caucasians and 12–23% in Asians [6]. This suggests that PPI activity may differ depending on the race and CYP2C19 genotype of the individual [7,8,9]. Systemic exposure of omeprazole after single and multiple administrations was higher in CYP2C19 PM than in CYP2C19 intermediate metabolizers (IM) and extensive metabolizers (EM) [10]. The means of the area under the curve from 0 to 12 h (AUC_0–12 h_) ratios in PM after a single administration were 3.76-fold higher than in IM (3652 vs. 972.5 μg × h/L) and 5.12-fold higher than in EM (3652 vs. 713.5 μg × h/L), whereas the mean AUC_0–12 h_ ratios in PM after multiple administrations were 1.76-fold higher than in IM (3669 vs. 2087 μg × h/L) and 2.14-fold higher than in EM (3669 vs. 1715 μg × h/L) [10].

Potassium-competitive acid blockers (P-CABs), such as revaprazan, vonoprazan, and tegoprazan, have several advantages over conventional PPIs. P-CABs exhibit weak or strong basic properties; thus, they are more effectively distributed in the parietal cell canaliculus than in the plasma [11]. Most PPIs, such as omeprazole, lansoprazole, pantoprazole, and rabeprazole, exhibit acid-labile properties; therefore, enteric-coated formulations have been widely used to prevent acidic degradation [12,13]. Meanwhile, P-CABs are relatively stable compared to conventional PPIs, even in the acidic gastric environment. Tegoprazan is a benzimidazole derivative that was developed by HK inno.N Corp in Korea. Compared with PPIs, tegoprazan does not need activation in an acidic environment; therefore, tegoprazan exhibits a faster onset than PPIs and has highly selective and reversible inhibitory properties against H^+^/K^+^-ATPase in parietal cells [14]. After the administration of tegoprazan at 50, 100, 200, or 400 mg doses to 32 healthy adult males, the median time of maximum concentration (T_max_) was 1 h (0.5–1.5 h), and the mean elimination half-life ranged from 3.65 to 5.39 h [15]. Compared to conventional PPIs, tegoprazan has a longer half-life and reversible binding properties, thus exhibiting a higher inhibition rate for nocturnal acid breakthrough. Tegoprazan demonstrated linear PK profiles; dose-dependent pharmacodynamic profiles were observed after tegoprazan administration over a dose range of 50–400 mg [16]. Tegoprazan is primarily metabolized by CYP3A4, and approximately 75% of the resulting metabolic fraction is subject to total hepatic clearance [17,18]. Therefore, tegoprazan has the advantage of being less affected by CYP2C19-mediated drug interactions than PPIs such as omeprazole, esomeprazole, and lansoprazole [16,19].

The physicochemical properties of tegoprazan may be affected by food or increased intragastric pH. The solubility of tegoprazan according to the patent is 0.7 mg/mL and 0.02 mg/mL at pH 3 and 6.8, respectively [20], and the predicted solubility in ChemAxon was 223 mg/mL at pH 1. Furthermore, tegoprazan exhibits a weak base effect (pKa = 5.1). These properties suggest that food or repeated administrations may influence PK profiles of tegoprazan. When omeprazole, pantoprazole, or rabeprazole were administered after a high-fat meal, PK parameters, such as T_max_, AUC, half-life, and maximum concentration (C_max_), tended to fluctuate significantly [21]. In contrast, PK parameters of vonoprazan were not significantly affected by food intake [22].

Physiologically based pharmacokinetic (PBPK) modeling and simulation can predict the dynamics of drug concentrations in humans and animals [23]. PK parameters, such as C_max_, AUC, and clearance (CL), can also be calculated using a traditional static model [24]. However, a static model requires a steady-state assumption and has the disadvantage of poor prediction of overall drug exposure [24]. The PBPK model incorporates in vitro (e.g., effective permeability, microsomal clearance, and unbound fraction in microsomes) and in vivo (e.g., blood-to-plasma partition coefficient, unbound fraction in plasma, and urinary/biliary clearance) parameters to predict systemic exposure and drug distribution in specific tissues. The PBPK model is considered useful for predicting formulation effects [25,26], PK profiles in special populations (e.g., pediatrics, renal/liver impairment, and pregnancy) [27], and drug–drug interactions (DDI) [28]. Recently, a PBPK model was used to predict PK profiles for a first-in-human study to determine optimal dosing regimens [29,30,31].

A recent study predicted the DDI of tegoprazan using a PBPK model [18] that was established using the advanced dissolution, absorption, and metabolism (ADAM) absorption model and the minimal PBPK distribution model. They evaluated the developed model using clinical data from single and repeated administrations of tegoprazan, and the model predicted the observed PK profiles well. When clarithromycin, which is used for *Helicobacter pylori* eradication therapy, was co-administered with tegoprazan, the established model could predict the potential for DDI. The predicted systemic exposure of tegoprazan increased by approximately three-fold when co-administered with the maximum recommended dose of clarithromycin or ketoconazole, compared with that when administered without a CYP3A4 modulator. Furthermore, when the CYP3A4 inducer rifampicin was administered with tegoprazan, systemic exposure of tegoprazan was predicted to decrease by up to 30%. Although the study predicted and validated CYP3A4-mediated DDI using this model, the effect of metabolites of tegoprazan and food were not considered.

The objectives of this study were (1) to establish a PBPK/pharmacodynamic (PD) model of tegoprazan and its major metabolite M1 using their physicochemical characteristics, in vitro experiments for absorption and metabolism, and clinical data; (2) to identify the factors that have a significant influence on the postprandial PK profiles of tegoprazan; and (3) to predict the change in intragastric pH following the administration of tegoprazan.

## 2. Materials and Methods

### 2.1. Reagents

Tegoprazan, tegoprazan-d_6_, and M1 were obtained from HK inno.N Corp. Caco-2 cells (passage no. 41–43) were purchased from the American Type Culture Collection (Rockville, MD, USA). Dulbecco’s modified Eagle’s medium (DMEM); fetal bovine serum (FBS); penicillin–streptomycin; 12-well Transwell plates; ultrapooled HLMs (150 donors, mixed gender); Supersome^®^ recombinant CYP3A4, 2C19, 2C9, 2C8, 2D6, and 2E1; and the nicotinamide adenine dinucleotide phosphate (NADPH)-generating system (1.3 mM NADP^+^, 3.3 mM glucose 6-phosphate, 3.3 mM MgCl_2_, and 0.4 unit/mL glucose-6-phosphate dehydrogenase) were purchased from Corning Life Sciences (Woburn, MA, USA). Hank’s balanced salt solution (HBSS), phosphate-buffered saline (PBS), non-essential amino acids, glucose 6-phosphate, chlorpropamide, Trizma^®^ base, Trizma^®^ hydrochloride, DMSO, and formic acid were purchased from Sigma-Aldrich Co. (St. Louis, MO, USA).

### 2.2. Assessment of Microsomal Metabolic Stability for Tegoprazan and M1

The NADPH-generating system, glucose 6-phosphate (3.3 mM), 0.1 M potassium phosphate (pH 7.4), and human liver microsomes (HLMs; 0.5 mg/mL) were combined and incubated at 37 °C for 5 min. The organic content in the reaction mixture was 0.01% DMSO and less than 1% acetonitrile. Then, 1 μM tegoprazan and M1 were added to the reaction tube (total volume: 160 μL) and incubated at 37 °C for 0, 5, 15, 30, and 60 min. To terminate the reaction, 320 μL of acetonitrile containing an internal standard (100 ng/mL carbamazepine) was added to each sample. All experiments were performed in triplicate. The samples were vortexed for 5 min and centrifuged at 3000 rpm at 4 °C for 10 min. The concentrations of tegoprazan and M1 were determined by liquid chromatography–tandem mass spectrometry (LC–MS/MS). The concentrations of tegoprazan and M1 at 0 min were used to assess metabolism by HLMs based on changes in drug concentration over time. The terminal slope of linear regression was used to calculate the intrinsic clearance (CL_int_) of tegoprazan and M1. For the in vitro HLM system, CL_int_ was calculated using Equation (1).
CL_int,mic_ (μL/min/mg protein) = k × V_incubation_/C_incubation_(1)
where V_incubation_ is the incubation volume and C_incubation_ is the concentration of the microsomal protein.

### 2.3. Caco-2 Permeability Assay

Caco-2 cells were cultured in DMEM supplemented with 20% FBS, 1% non-essential amino acids, and 1% penicillin-streptomycin. Caco-2 cells were seeded onto collagen-coated 12-Transwell membranes at a density of 5 × 10^5^ cells/mL and incubated under optimal conditions (maintained at 37 °C in a humidified atmosphere with 5% CO_2_/95% air) for 21 days. The culture medium was replaced every second day. For the experiments, after the growth medium was removed, the attached cells were washed with pre-warmed HBSS (pH 7.4) and pre-incubated with HBSS for 20 min at 37 °C. The integrity of the cell monolayers was evaluated before and after the permeability experiments by measuring transepithelial electrical resistance (TEER) values using an epithelial volt/ohm meter (World Precision Instruments, Sarasota, FL, USA). Caco-2 cells with TEER values higher than 300 Ω∙cm^2^ were used for permeability experiments, and alterations in TEER values were less than 10% in all cases (i.e., 417.3 ± 6.3 Ω∙cm^2^ and 400.8 ± 3.5 Ω∙cm^2^, before and after the permeability experiments, respectively).

To determine the apical to basal (A to B) permeability of tegoprazan, 0.5 mL of HBSS containing 10 µM tegoprazan and 1.5 mL of fresh HBSS were added to the apical and basal side of the insert, respectively. The insert was transferred to a new well that contained 1.5 mL of fresh HBSS every 15 min, for 1 h. Aliquots (0.3 mL) on the basal side were transferred to fresh tubes and stored at −80 °C until further analysis. To investigate B-to-A permeability, 1.5 mL of HBSS containing 10 µM tegoprazan and 0.5 mL of fresh HBSS were added to the basal and apical sides of the insert, respectively. Aliquots (0.3 mL) on the apical side were transferred to fresh tubes, and 0.3 mL of pre-warmed fresh HBSS was replaced every 15 min for 1 h. The transport study was performed for 1 h based on the linearity of the cumulative transported tegoprazan amount versus transport time. The cumulative transported amount was calculated as the sum of transported amounts for 0–15 min, 15–30 min, 30–45 min, and 45–60 min, and the transport rate of tegoprazan was the slope of the regression line derived from the cumulative transported amount versus time plot. The apparent permeability (P_app_) was calculated from the following Equation (2):(2)$$P_{\mathrm{app}}\left(10^{-6}\mathrm{cm}/s\right)=\frac{\mathrm{transport}\;\mathrm{rate}\;\left(\mathrm{nmol}/\min\right)}{\mathrm{concentration}\;\left(\mu M\right)\times \;\mathrm{area}\left({\mathrm{cm}}^{2}\right)\times 60\;s}$$

### 2.4. Evaluation of Intrinsic Clearance Using Recombinant CYPs (rCYPs)

The elimination profiles of tegoprazan and M1 were assessed using rCYPs. All the assay procedures were performed according to the manufacturer’s instructions. Master mix (1.399 mL), containing the NADPH-generating system, 3.3 mM of glucose 6-phosphate, PBS buffer, deionized water, and each rCYP (25 pmol/μL of CYP2C8, 2C19, and 2E1 and 50 pmol/μL of CYP3A4, 2C9, and 2D6), was added to a microtube and pre-incubated at 37 °C for 5 min. Next, 1 μM tegoprazan and M1 were added and incubated at 37 °C for 0, 5, 10, 20, 30, 45, and 60 min. A total of 100 μL of the sample, incubated for predefined duration, was transferred to a fresh e-tube. Next, 200 μL of acetonitrile containing tegoprazan-d_6_ (0.5 μM) was added, vortexed for 5 min, and centrifuged at 16,200× *g* and 4 °C for 10 min. The concentrations of the remaining tegoprazan and M1 were determined by LC–MS/MS. All experiments were performed in duplicate. CL_int,rCYP_ levels were assessed using the results of the disappearance test. The terminal slope of the linear regression was used to determine the CL_int_ for tegoprazan. The CL_int_ resulting from rCYPs was calculated using Equation (3):(3)$${\mathrm{CL}}_{\mathrm{int},\mathrm{rCYP}}(\mu L/\min/\mathrm{pmol}\;\mathrm{CYP})=\frac{0.693}{t_{1/2}}\times \frac{1400}{\mathrm{concentration}\;\mathrm{of}\;\mathrm{CYP}}$$

### 2.5. Determination of Tegoprazan and M1 Concentration in In Vitro Samples

Caco-2 and rCYP samples were quantified using LC-MS/MS (UPLC^®^ I-class coupled with TQ-MS (Waters, Milford, MA, USA). The mobile phase comprised (a) 0.1% formic acid and (b) acetonitrile containing 0.1% formic acid. Gradient elution at a flow rate of 0.3 mL/min was applied. Tegoprazan, M1, and the internal standard (tegoprazan-d_6_) were optimized in electrospray ionization positive mode. Acetonitrile with 0.1 μM of tegoprazan-d_6_ was added to the collected Caco-2 or rCYP samples. Then, processed samples were centrifuged at 13,000× *g* for 20 min. After centrifugation, 100 μL of supernatant was transferred to an LC-MS vial and analyzed using LC-MS/MS. The analytical method was validated based on the *Bioanalytical Method Validation Guidance for Industry* published by the US Food and Drug Administration and the *Guideline on Bioanalytical Method Validation* published by the Korean Ministry of Food and Drug Safety (MFDS) [32,33]. Linearity was in the range of 0.1–20 µM, and the inter-batch accuracy and precision were in the ranges of 91.8–101.3% and 2.3–8.9%, respectively. The stability of tegoprazan was evaluated, and it satisfied the acceptance criteria (coefficient of variances was within 15%) under variable conditions including reinjection, processed sample stability (PSS), and short-term stability for the stock and sample.

### 2.6. Clinical PK Dataset

Plasma tegoprazan and M1 concentration–time profiles after a single oral administration were obtained from a previous comparative pharmacokinetic study comparing two formulations of tegoprazan in healthy Korean male subjects (IRB number: CUH 2015-03-007) [34]. This study was designed as a randomized, open-label, single-dose, two-sequence, and two-period crossover study. A total of 12 volunteers participated, and the mean ± standard deviation of age, height, and weight were 23.9 ± 1.3 years, 173.1 ± 7.6 cm, and 68.4 ± 8.2 kg, respectively. All subjects were administered a single tegoprazan film-coated 100 mg tablet orally of formulation type 1 or 2 (HK inno.N Corp., Seoul, Korea), and pharmacokinetic profiles were analyzed. For pharmacokinetic analysis, 8 mL of blood was collected to measure the plasma tegoprazan and M1 concentrations during each period at the following time points: pre-dose (0 h) and 0.25, 0.5, 1, 1.5, 2, 3, 4, 6, 8, 12, 24, and 48 h post-dose. Plasma tegoprazan and M1 concentrations were measured by liquid chromatography–tandem mass spectrometry (LC–MS/MS). Individual concentration–time profiles were used to develop and verify tegoprazan and M1 PBPK models. The PK parameters were calculated using a non-compartmental analysis (NCA) performed by Phoenix^®^ WinNonlin^®^ 6.2 (Certara, Princeton, NJ, USA).

### 2.7. PBPK Model Development

The tegoprazan and M1 PBPK models were developed using SimCYP^®^ simulator, version 20 (Certara, Princeton, NJ, USA). Compound libraries were incorporated into an in vitro dataset, including the microsomal clearance of tegoprazan and M1. Observed PK parameters were obtained from the aforementioned clinical studies (Table 1 and Table 2).

The absorption of tegoprazan was applied to the advanced dissolution, absorption, and metabolism (ADAM) model, and the diffusion layer model (DLM) built into SimCYP^®^ was applied to the considered formulation and pH-dependent solubility of tegoprazan. The permeability of tegoprazan was determined by a Caco-2 permeability assay and optimized by parameter estimation. Caco-2 permeability was optimized such that the predicted concentration–time profiles were fitted to clinical study data. The metabolite was generated by the metabolism of the parent drug; therefore, absorption was not considered.

The tissue distribution in the PBPK model was predicted by the distribution model (e.g., Rodger and Rowland method, Poulin and Theil method) and the tissue:plasma partition coefficient (Kp) scalar for specific organ or global values. Predicted systemic distribution was expressed as the volume of distribution in a steady state (V_ss_). The V_d_ values of the clinical study were applied to the initial model of tegoprazan, and parameter estimation was used to determine the optimized Kp scalar fitted to the observed data. The Rodger and Rowland method (method 2) was applied to consider the weak base property and distribution of the neutral and ionic forms [35]. The minimal PBPK model was applied to the M1 distribution module to reduce the complexity. The volume of a single adjusting compartment (V_sac_), non-physiological compartment, input rate to SAC (k_in_), and elimination rate from SAC (k_out_) were estimated using the parameter estimation module within SimCYP^®^.

The elimination profiles of tegoprazan and M1 were obtained from in vitro, in vivo, and in silico predictions. Optimized in-house intrinsic clearance for CYPs (CL_int,rCYPs_), and human liver microsomal and cytosolic (HLM and HLC) clearance were applied to the elimination module of the tegoprazan model. Observed or estimated renal and biliary clearance were applied to the tegoprazan and M1 models [36]. Whole-organ metabolic clearance, using the optimized value for the in vitro assay results, was applied to the M1 model.

**Table 1 pharmaceutics-14-01298-t001:** The input parameters for the tegoprazan PBPK model.

Parameters	Initial Value	Final Value	Source
*Phys-chem properties*			
Molecular weight (g/mol)	387.38	387.38	Predicted using ChemAxon
LogP	2.323	2.323	Predicted using ChemAxon
Compound type	Monoprotic base	Monoprotic base	
pKa	5.1	5.1	[15]
B/P	0.792	0.792	Predicted in SimCYP^®^
f_u_	0.087	0.087	[36]
*Absorption*			
Absorption model	ADAM	ADAM	
Permeability assay	caco-2	caco-2	
P_app,caco-2_ (×10^−4^ cm/s)	0.1253	1.00	Optimized value using experimental value and parameter estimation
*Distribution*			
Distribution model	Full PBPK	Full PBPK	
V_ss_ (L/kg)	1.55	1.0	Adjusted by Kp scalar
Prediction model	Method 2 (Rodgers and Rowland model)	Method 2 (Rodgers and Rowland model)	
*Elimination*			
Clearance type	Enzyme kinetics	Enzyme kinetics	
CL_int, CYP3A4_ (μL/min/pmol)	0.855	1.920	Optimized value using experimental data and parameter estimation
CL_int, CYP2C19_ (μL/min/pmol)	0.614	0.710	Optimized value using experimental data and parameter estimation
CL_int, CYP2C8_ (μL/min/pmol)	0.060	0.060	Experimental data
CL_int, CYP2C9_ (μL/min/pmol)	0.140	0.140	Experimental data
CL_int, CYP2D6_ (μL/min/pmol)	0.020	0.020	Experimental data
CL_int, CYP2E1_ (μL/min/pmol)	0.030	0.030	Experimental data
Additional HLM CL_int_ (μL/min/mg protein)	15.96	15.96	Experimental data
Biliary CL_int_ (µL/min/10^6^)	-	1.290	Parameter estimation
CL_R_ (L/h)	1.1	1.1	[36]
Additional systemic clearance (L/h)	-	1.43	Parameter estimation

ADAM, advanced dissolution, absorption, and metabolism; B/P, blood-to-plasma ratio; f_u_, unbound fraction; P_app_,_caco-2_, apparent permeability of caco-2; V_ss_, volume of distribution at steady state; CL_int_, intrinsic clearance; CL_R_, renal clearance.

**Table 2 pharmaceutics-14-01298-t002:** The input parameters for the tegoprzan-M1 PBPK model.

Parameters	Initial Value	Input Value	Source
*Phys-chem properties*			
Molecular weight (g/mol)	373.36	373.36	Predicted using ChemAxon
LogP	2.1	2.1	Predicted using ChemAxon
Compound type	Monoprotic base	Monoprotic base	
pKa	5.35	5.35	Predicted using ChemAxon
B/P	1.116	1.116	Predicted in SimCYP^®^
f_u_	0.257	0.257	Predicted in SimCYP^®^
*Distribution*			
Distribution model	Minimal PBPK	Minimal PBPK	
V_ss_ (L/kg)	-	1.72	Parameter estimation
k_in_ (h^−1^)	-	40	Parameter estimation
k_out_ (h^−1^)	-	7.76	Parameter estimation
V_sac_ (L/kg)	-	1.23	Parameter estimation
Prediction model	Method 2 (Rodgers and Rowland model)	Method 2 (Rodgers and Rowland model)	
*Elimination*			
Clearance type	WOMC	WOMC	
CL_int,HLM_ (μL/min/mg protein)	2.353	2.353	Experimental data
CL_int,HLC_ (μL/min/mg protein)	-	3.15	Parameter estimation
CL_R_ (L/h)	1.1	1.1	Assumed that same with parent
Additional systemic clearance (L/h)	-	1.44	Parameter estimation

f_u_: unbound fraction in plasma; V_ss_: volume of distribution in steady state; k_in_: input rate constant for single adjusting compartment; k_out_: elimination rate constant for single adjusting compartment; V_sac_: volume of single adjusting compartment; WOMC: whole-organ metabolic clearance; HLC: human liver cytosol; HLM: human liver microsomes; CL_int_: intrinsic clearance; CL_R_: renal clearance.

### 2.8. Model Verification

The virtual dosing regimen was set to a single oral dose of 100 mg tegoprazan, to match that of the observed data. The simulation conditions were set for 100 virtual subjects who participated in 10 virtual clinical studies (1000 volunteers). The age range was 20 to 40 years, and the male-to-female ratio was 1:1. The predicted and observed PK parameters were compared to evaluate the model prediction results. The AUC_0–24 h_, maximum plasma concentration (C_max_), time required to reach the C_max_ (T_max_), and CL for tegoprazan and AUC_0–48 h_ and C_max_ for M1 are presented as results, and the predicted/observed ratios for AUC, C_max_, and CL were calculated to estimate the predictive power of the model. If the predicted vs. observed ratio and its 95% CI were within the 30% range (0.7–1.3), the model was considered to fit well.

### 2.9. PBPK Model Application

#### 2.9.1. Prediction of Systemic Exposure to Tegoprazan and M1 after Repeated Administrations

Using the extracted plasma tegoprazan concentration–time profile reported in the literature [15], the predicted parameters (AUC_0–24 h_ and C_max_) under conditions of repeated administrations for 7 days (100 mg of tegoprazan q.d.) were compared with the results of the clinical studies.

#### 2.9.2. Prediction and Comparison of Postprandial PK Profiles Using the Developed Model

Postprandial PK profiles were predicted using an established model. Concentration–time profiles were obtained following a single oral dose (50 mg) of tegoprazan in the fasting state or 30 min after a high-fat diet. The predicted AUC_last_, C_max_, and T_max_ values were compared with those of a previous clinical study [16].

### 2.10. Incorporation of a PD Model to Predict Tegoprazan-Induced Gastric pH Changes

To simulate the change in gastric acid pH after the administration of tegoprazan, an intragastric PD model following the administration of tegoprazan was incorporated into the developed PBPK model. The trend of the 24-h gastric acid pH baseline was described using a sixth order Fourier series, and the sigmoid E_max_ model was applied to the indirect response model based on the baseline intragastric pH model used to predict gastric acid pH alterations after tegoprazan administration. The equations for the indirect response-sigmoid E_max_ model with the baseline intragastric pH model were as follows:
(4)$$\mathrm{Effect}=\mathrm{baseline}\mathrm{pH}\times (1+\frac{E_{\max}\times {C_{p}}^{\gamma }}{{{\mathrm{EC}}_{50}}^{\gamma }+{C_{p}}^{\gamma }})$$
(5)$$\mathrm{Base}=a_{0}+\sum_{i=1}^{6}a_{i}\cos\left(\mathrm{iwx}\right)+b_{i}\sin\left(\mathrm{iwx}\right)$$
(6)$$\frac{\mathrm{dR}}{\mathrm{dt}}=k_{\mathrm{in}}\times \left(\mathrm{Effect}+\mathrm{Base}\right)-k_{\mathrm{out}}\times R$$
where Base is the baseline intragastric pH model; $a_{0}$, $a_{i}$ and $b_{i}$ are the Fourier coefficients; i is an integer sequence; w is the fundamental frequency; x is time; E_max_ is the maximum effect; C_p_ is plasma tegoprazan concentration; EC_50_ is the concentration at half of the maximum effect; k_in_ is the predicted plasma tegoprazan concentration; k_out_ is the output rate constant; R is the response; and γ is the Hill coefficient. Furthermore, the predicted PD response was reflected into the individual gastric acid profiles in the population library used in the PBPK simulation to apply the gastric acid pH-lowering effect (Figure 1). A PD model with gastric pH feedback function was generated using a custom Lua script within the SimCYP^®^ simulator.

The established PBPK/PD model was evaluated using observed plasma tegoprazan concentrations and gastric acid pH-time profiles after a single administration of 50 mg tegoprazan. This model was used to predict the PK and gastric acid pH profiles after repeated administrations of 100 mg tegoprazan once daily for 7 days. The predicted PD response was evaluated from the extracted gastric acid pH profiles following the administration of tegoprazan [15]. The results of the PD model were evaluated at a holding rate of pH ≥ 4, which is the surrogate endpoint of the GERD healing rate [37,38].

## 3. Results

### 3.1. Caco-2 Permeability Assay

The mean ± standard deviation of apical to basolateral (A to B) permeability was 12.53 ± 1.32 × 10^−6^ cm/s (Table 3). The recovery calculated through an analysis of the results and the P_app_ value satisfied the range of ±20%; therefore, we confirmed that this was not an error during the experimental processes. Moreover, since the efflux ratio was lower than 2, the effect of transport via p-glycoprotein is expected to be insignificant. Mean caco-2 permeability was used to develop the initial tegoprazan PBPK model and subsequent model improvement.

### 3.2. Assessment of Microsomal Metabolic Stability for Tegoprazan and M1

In the control sample without NADPH, the percentages of tegoprazan and M1 remaining after incubation were 99.54% ± 5.32% and 102.2% ± 6.52%, respectively, indicating that the disappearance of tegoprazan and M1 primarily occurred via hepatic CYPs. The HLM intrinsic clearance (CL_int,HLM_) of tegoprazan and M1 was 15.96 and 2.44 µL/min/mg protein, respectively (Figure 2). These values were applied to the PBPK model.

### 3.3. Evaluation of Intrinsic Clearance Using Recombinant CYPs

CL_int,3A4_ and CL_int,2C19_ were 0.86 and 0.61 µL/min/pmol recombinant CYP, respectively (Appendix A). The CL_int_ values for CYP2C8, 2C9, 2D6, and 2E1 were 0.06, 0.14, 0.02, and 0.03 µL/min/pmol of rCYP, respectively (Appendix A). CL_int,CYP_ for tegoprazan were applied to enzyme kinetics in the elimination module.

### 3.4. Model Verification

The final model was used to predict the PK profiles of tegoprazan and M1. Most of the observed mean plasma concentration–time profiles were included in the 5th and 95th percentiles of the predicted mean plasma concentration–time profiles (Figure 3). The ratios of predicted vs. observed AUC_last_, C_max_, and CL, and 95% confidence interval values were included in the 30% range of the mean ratio (Table 4).

### 3.5. Model Application

#### 3.5.1. Prediction of Systemic Exposure to Tegoprazan and M1 after Repeated Administrations

Since it is common for gastric acid pH to increase throughout repeated tegoprazan administrations for gastroesophageal reflux disease, gastric ulcer, and *Helicobacter pylori* eradication, which are the primary indications for tegoprazan [39], the plasma concentration–time profiles were predicted for repeated administrations using the established model. The predicted mean plasma tegoprazan concentration–time profiles at a dose of 100 mg were fitted to the observed profiles (Figure 4). However, the developed model overpredicted the C_max_ on Day 7. The predicted C_max_ of tegoprazan was 1.46-fold higher than the observed C_max_ (predicted C_max_ 1257.8 ± 250.5 ng/mL vs. observed C_max_ 845.2 ± 40.7 ng/mL).

#### 3.5.2. Prediction and Comparison of Postprandial PK Profiles Using the Developed Model

The ratios of predicted vs. observed tegoprazan AUC_0–24 h_ and C_max_ values were within the 30% range of the mean ratio, which is consistent with previously reported results (Table 5) [16]. For M1, the ratio of the predicted vs. observed AUC_0–48 h_ ratio was within the 30% range of the mean ratio; however, the predicted vs observed C_max_ ratio exceeded the 30% range. For drug administered after a high-fat diet, the predicted mean C_max_ of tegoprazan decreased by 31.3% (fasted state: 684.9 ng/mL; at 30 min after a high-fat meal: 470.7 ng/mL) and the predicted median T_max_ was delayed from 0.6 h (0.3–1.3) to 1.05 h (0.3–3.5). The predicted AUC_0–24 h_ of tegoprazan was not significantly different from that in the fasted state. The predicted mean AUC_0–48 h_ and C_max_ of M1 did not show large differences compared to fasting, and the predicted median T_max_ of M1 was delayed from 5.8 h (1.5–23) to 7.5 h (2.3–24).

### 3.6. Incorporation of a PD Model to Predict Tegoprazan-Induced Gastric pH Changes

The established PBPK/PD model successfully predicted plasma tegoprazan concentrations and gastric acid pH-time profiles after a single administration of 50 mg tegoprazan (Figure 5).

Plasma tegoprazan concentrations and gastric acid pH–time profiles after repeated administrations of 100 mg tegoprazan once daily for 7 days were predicted using the established PBPK/PD model. Most of the observed concentration profiles on Day 1 and 7 fit well into the 5th and 95th percentile intervals of the predicted mean concentration (Figure 6a,c). When the PD response was not applied, the intragastric pH was fixed at 1.49, which was the default baseline pH (Figure 6b). However, when the PD response with feedback function was applied, the intragastric pH changed over time (Figure 6d). The observed gastric acid pH profiles were also well fitted to the predicted gastric acid pH–time profiles (Figure 6d).

When evaluating the percent holding rate at pH ≥ 4 for 24 h, which is a PD parameter, the holding rate after a single administration of 50 mg tegoprazan or repeated administrations of 100 mg tegoprazan once daily for 7 days matched the results of the clinical studies (Table 6).

## 4. Discussion

A PBPK/PD model for tegoprazan and M1 was successfully established. In vitro permeability and metabolism data and clinical study results were used to develop and evaluate the model. Plasma tegoprazan and M1 concentration–time profiles and gastric acid pH–time profiles after single or repeated administrations were predicted using the developed model. The ratios of predicted vs. observed AUC_0–24 h_, C_max_, and CL (95% confidence intervals) values for tegoprazan were 0.90 (0.85–0.96), 0.88 (0.80–1.00), and 1.25 (1.17–1.34), respectively. The predicted results of the M1 model revealed that the arithmetic mean ratios (95% confidence interval) of the predicted AUC_0–48 h_ and C_max_ to the observed plasma concentration–time profile were 0.88 (0.84–0.94) and 1.01 (0.95–1.08), respectively (Table 4). According to the simulation results, the developed model accurately predicted the PK profiles of tegoprazan and M1. The PBPK model was applied to predict PK profiles following repeated administrations of tegoprazan, postprandial PK profiles, and alteration of gastric acid pH profiles after tegoprazan administration. When the clinical study and predicted results were compared, good prediction results were evident under all conditions. Thus, we confirmed that the developed PBPK/PD model had sufficient predictive power.

The elimination profiles of tegoprazan were reflected in the PBPK model through a metabolism assay using recombinant CYPs. Based on previous reports, tegoprazan is primarily metabolized by CYP3A4 [18,40]. According to the information leaflet for K-CAB^®^ tablets (tegoprazan 50 mg), although the metabolism of tegoprazan is decreased in the presence of ketoconazole, other CYP1A2, 2C9, 2C19, and 2D6 inhibitors did not significantly decrease the in vitro metabolism of tegoprazan (exact value not presented) [36]. When comparing the results of applying only the intrinsic clearance of CYP3A4 and 2C19 to the elimination module of the tegoprazan model with those of the intrinsic clearance of CYP2C8, 2C9, 2D6, and 2E1 together with CYP3A4 and 2C19, no significant differences were observed (Appendix A). Both CYP3A4- and 2C19-mediated metabolism of tegoprazan are thought to be major metabolic pathways. Based on these results, the CYP3A4- and 2C19-mediated metabolism of tegoprazan is likely to be a major metabolic enzyme involved in tegoprazan metabolism.

The PBPK/PD model developed in this study successfully predicted the PK profile of M1, the major metabolite. In a preclinical study, M1 showed reversible inhibitory potential against porcine H^+^/K^+^-ATPase, with 10-fold less potency than tegoprazan [41]. The experimental half maximal inhibitory concentration (IC_50_) values for tegoprazan and M1 were 0.53 μM and 6.19 μM, respectively [40,41]. M1 is also expected to exhibit an acid suppression effect based on preclinical studies and in vitro assay results; however, no studies have evaluated the efficacy of M1 in humans. In the present study, a PBPK model for M1 was developed using minimal PBPK distribution and clearance profiles in HLM to evaluate PK profiles. The developed M1 model may contribute to the mechanistic interpretation of DDI in the PK profiles of M1 and tegoprazan.

Integrating the PBPK model with an intragastric PD model, gastric pH changes after tegoprazan administration were obtained. To apply the circadian rhythm of intragastric pH, a Fourier series was fitted to the baseline pH profile and the profile was applied to the basic structure of the PD model. The indirect response-sigmoid E_max_ model with the baseline intragastric pH model and gastric acid pH feedback function of the ADAM model was integrated into the PBPK model. The PD response was successfully predicted with the changes of intragastric pH in virtual subjects after tegoprazan administration. Using the integrated model, PD responses under various clinical settings could be predicted.

The developed PBPK/PD model may be used to evaluate the effect of tegoprazan-induced intragastric pH increase on the pharmacokinetics of drugs co-administered with tegoprazan. The predicted results using the established model indicated that the observed plasma tegoprazan concentration and gastric acid pH–time profiles after a single administration of 50 mg tegoprazan and repeated administrations of 100 mg tegoprazan once daily for 7 days were well fitted to the prediction profiles. The percentage holding rates of pH ≥ 4 for 24 h after a single administration of 50 mg tegoprazan or repeated administrations of 100 mg tegoprazan for 7 days were similar to those of the clinical study. The percent holding rate at pH ≥ 4 for 24 h after a single administration of 50 mg tegoprazan was predicted to be approximately 49.6%. The percent holding rates at pH ≥ 4 for 24 h after repeated administrations of 100 mg tegoprazan once daily for 7 days were predicted to be approximately 60.3% on Day 1 and Day 7.

According to solubility profiles, repeated administrations of tegoprazan may affect its absorption. The pKa of tegoprazan is 5.1, which is indicative of a weak base [40,41]. Due to this physicochemical property, tegoprazan may be specifically highly distributed in the parietal cells of the stomach, where the pH is 1–2. This indicates that tegoprazan has the potential to effectively suppress gastric acid secretion [17,42]. In the clinical study, after repeated administrations of 100 mg tegoprazan once daily, the observed C_max_ at Day 7 decreased by approximately 40.2% compared with Day 1 (Day 1: 1413.3 ± 24.7 ng/mL; Day 7: 845.2 ± 40.7 ng/mL) [15]. When 100 mg of tegoprazan was repeatedly administered to 10 healthy Chinese subjects, AUC and CL on Day 7 of administration did not show a significant difference compared with Day 1 after administration of tegoprazan; however, V_d_/F increased significantly on Day 7 compared with that on Day 1 [39], whereas the predicted C_max_ at Day 7 increased by approximately 1.84% compared with Day 1 (Day 1: 1235.1 ± 241.9 ng/mL; Day 7: 1257.8 ± 250.5 ng/mL). Based on these results, it was suggested that repeated administrations of tegoprazan might increase intragastric pH, thereby reducing the absorption of tegoprazan itself.

The decrease in C_max_ after repeated administrations of tegoprazan appears to be influenced not only by the decrease in acidity of gastric juice but also by other physiological or drug-specific factors. The developed model also overpredicted the C_max_ on Day 7 compared with the observed profiles after repeated administrations (observed value: 845.2 ± 40.7 ng/mL; predicted value: 1257.8 ± 250.5 ng/mL). Based on the predicted results using the PBPK/PD model, gastric acid pH increased after taking tegoprazan; however, the systemic exposure or C_max_ of tegoprazan was barely affected (C_max_ of Day 7 when gastric acid pH was fixed at 1.49: 1266.1 ± 251.2 ng/mL; C_max_ of Day 7 when dynamic gastric acid pH by PD model: 1257.8 ± 250.5 ng/mL). In a previous study, it was estimated that the cause of increase in the volume of distribution following repeated administrations of tegoprazan was not only due to ionized molecules by gastric acid accumulated in the acidic secretory canaliculi of parietal cells, but also because the ionized form has poor cell membrane permeability [39]. The volume of distribution could be increased after repeated administrations of tegoprazan. Moreover, dissolution-related parameters, such as supersaturation ratio and precipitation rate constant, might influence the decrease in C_max_ values. Tegoprazan showed high permeability (experimental Caco-2 permeability: 12.53 ± 1.32 × 10^−6^ cm/s) and low solubility (0.02 mg/mL in pH 6.8), and it was considered a biopharmaceutics classification system (BCS) class II drug. In a PBPK study for BCS class II drugs, the colonic absorption, bile micelle solubilization, and unbound fraction in gut enterocytes (fu_gut_) were identified as important factors for the absorption process. In another PBPK study using a compound with pH-dependent solubility, particle size was considered a significant factor for drug exposure [42]. Although DLM was applied in the absorption module to reflect the effect of the formulation in the current model, the default values provided in SimCYP^®^ for formulation-related parameters, such as particle size, radius, density, and supersaturation ratio, were applied owing to a lack of information. This might lead to some discrepancies in the absorption process and related parameters.

The differences in residence time resulting from the diet, along with pH changes in the gastrointestinal tract, were supposed to affect the postprandial PK profiles of tegoprazan. According to a study that assessed differences in tegoprazan pharmacokinetics when administered in the fasting state, 30 min before a high-fat meal and 30 min after a high-fat meal, the C_max_ and T_max_ of tegoprazan were affected by food consumption, although systemic exposure of tegoprazan did not change. The C_max_ for 30 min after a high-fat meal decreased to 38.7% compared with that after a fasting state (fasting state: 803 μg/L; administered at 30 min after high-fat meal: 492 μg/L). The median T_max_ values were 1 h in the fasting state, 0.48 h at 30 min before a high-fat meal, and 3 h at 30 min after a high-fat meal [16]. These results indicated that the difference in postprandial PK profiles resulted from the physicochemical characteristics of tegoprazan and the time delay in reaching the small intestine [16]. Postprandial PK profiles of tegoprazan were predicted using an established model. When a high-fat diet was applied to the simulation, T_max_ was delayed and C_max_ was decreased, which is consistent with the results of the clinical study (Table 5). The predicted absorbed fraction in the duodenum upon administration of tegoprazan at 30 min after a high-fat meal was increased by approximately 72.7%, however, it decreased in other segments, compared with that in the group administered while fasting (Appendix A). The predicted surface solubility increased by 14.8% and 40.3% in the stomach and duodenum, respectively, and there was a slight decrease or no difference in solubility in the remaining segments in this group compared to that in a fasted state (Appendix A). The predicted mean residence time in the stomach upon administration of tegoprazan at 30 min after a high-fat meal was increased by approximately 170.4% compared to that administered while fasting (Appendix A). The predicted mean gastric residence time was delayed; therefore, it was assumed that tegoprazan was more distributed to parietal cells in the fed state than in the fasting state. Increases in the intragastric pH after tegoprazan administration or food intake are thought to be the major cause of a decrease in the C_max_ of tegoprazan. However, physiological factors (i.e., gastric and small intestine mean gastric residence times) and formulation-related parameters (i.e., particle size, bile micelle-mediated solubility parameters) can also affect postprandial PK profiles.

The limitation of this study is that the food effect was not considered in the PD model. In the SimCYP^®^ simulator, physiological factors in the gastrointestinal tract, such as intragastric/gastrointestinal (GI) tract pH and gastric emptying time, are applied differently in the fasted and fed states. If the drug response predicted by the PD model is applied to the intragastric pH of the virtual population, the dynamics of intragastric pH in the fed state predicted by the internal algorithm may not be reflected. If additional data on the actual conditions are available, such as the food effect or impact of the co-administered drug, and the circadian rhythm associated with intragastric pH can be reflected, the PBPK/PD model may predict a more realistic intragastric pH profile. The current model was built with extensive parameter optimization. When in vitro Caco-2 permeability and the intrinsic clearance of rCYPs were applied to the model, predicted concentration–time profiles were different from those of clinical study data. Caco-2 permeability and rCYP intrinsic clearance were estimated using parameter estimation, and the optimized value was fitted to the observed clinical study data. Although the final model was improved with extensive parameter optimization, it was validated using separate clinical study data that were not used for model development and model improvement. Therefore, the tegoprazan–M1 PBPK model developed in this study is considered a reasonable model.

## 5. Conclusions

In conclusion, a PBPK/PD model for tegoprazan and its metabolite M1 was developed by incorporating experimental in vitro metabolism and absorption profiles and clinical data. This model was successfully verified using the observed data following a single administration of tegoprazan. Moreover, the developed model fit the observed PK profiles well after repeated administrations. With respect to predicting postprandial PK profiles, T_max_ was delayed, and C_max_ was decreased. Although the absorbed fraction of tegoprazan increased in some GI tract segments in the postprandial state, it was expected that the delay in gastric and small intestine mean residence time would have a greater effect. The incorporated PD model provides a basis for evaluating changes in intragastric pH following tegoprazan administration. These PBPK/PD approaches may provide fundamental concepts for interpreting postprandial PK profiles and the impact of intragastric pH alterations on the co-administration of drugs with tegoprazan.

## Figures and Tables

**Figure 1 pharmaceutics-14-01298-f001:**
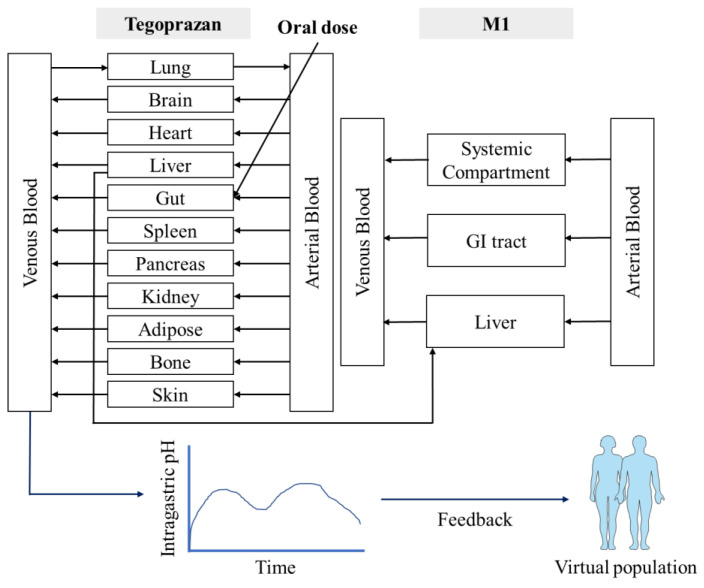
Structure of the tegoprazan-M1 PBPK/PD model.

**Figure 2 pharmaceutics-14-01298-f002:**
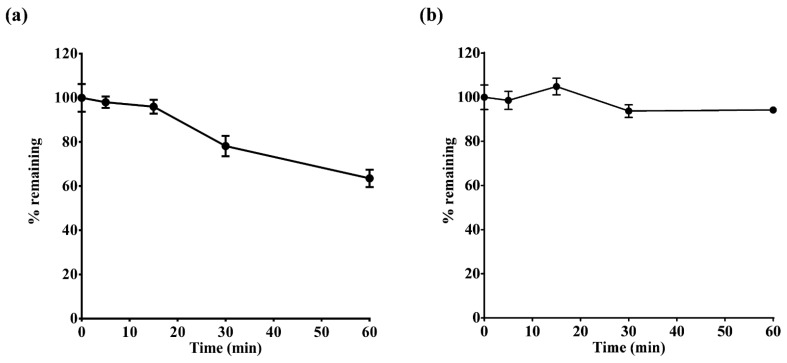
Percent remaining of (**a**) tegoprazan and (**b**) M1 following incubation with human liver microsomes (HLMs) and nicotinamide adenine dinucleotide phosphate (NADPH) cofactor.

**Figure 3 pharmaceutics-14-01298-f003:**
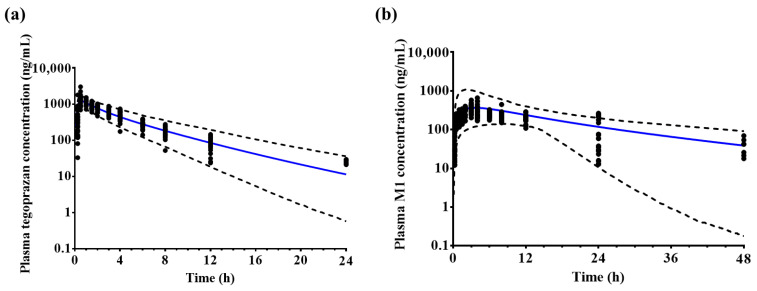
Observed (black dot, n = 12) and predicted (n = 1000, blue solid line) plasma (**a**) tegoprazan and (**b**) M1 concentration–time profiles after a single administration of 100 mg tegoprazan. Black dotted lines represent the predicted 95th and 5th percentiles.

**Figure 4 pharmaceutics-14-01298-f004:**
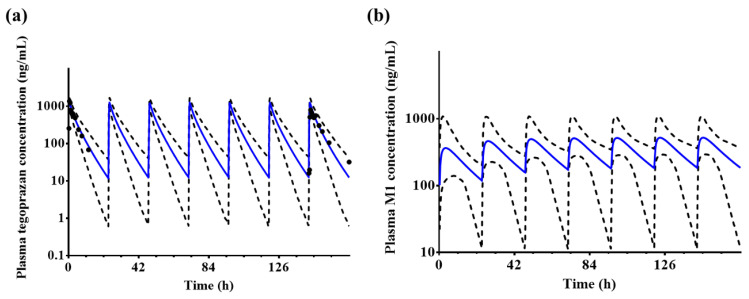
Predicted plasma (**a**) tegoprazan and (**b**) M1 concentration–time profiles after the repeated administration of 100 mg tegoprazan once daily for 7 days. Blue solid lines and black dotted lines represent the mean predicted plasma concentrations and the 95th and 5th percentiles, respectively. Black dots represent the observed tegoprazan concentration profiles (n = 6). Only the predicted profiles are presented for M1.

**Figure 5 pharmaceutics-14-01298-f005:**
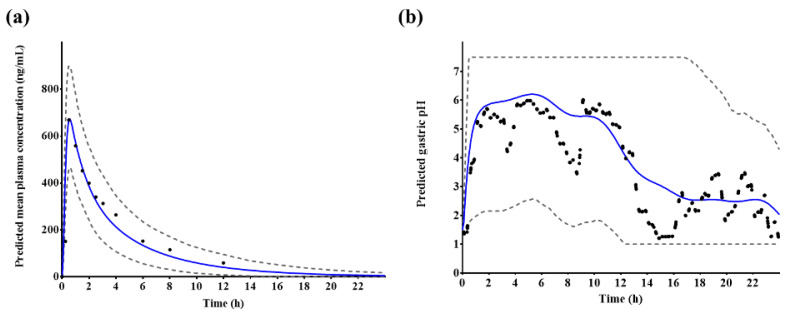
Predicted (**a**) plasma tegoprazan concentration and (**b**) gastric acid pH–time profiles after a single administrations of 50 mg tegoprazan. Blue solid lines represent the predicted mean values, gray dotted lines represent the predicted 5th and 95th percentile, and black dots represent the observed plasma tegoprazan concentration or gastric acid pH profiles, respectively.

**Figure 6 pharmaceutics-14-01298-f006:**
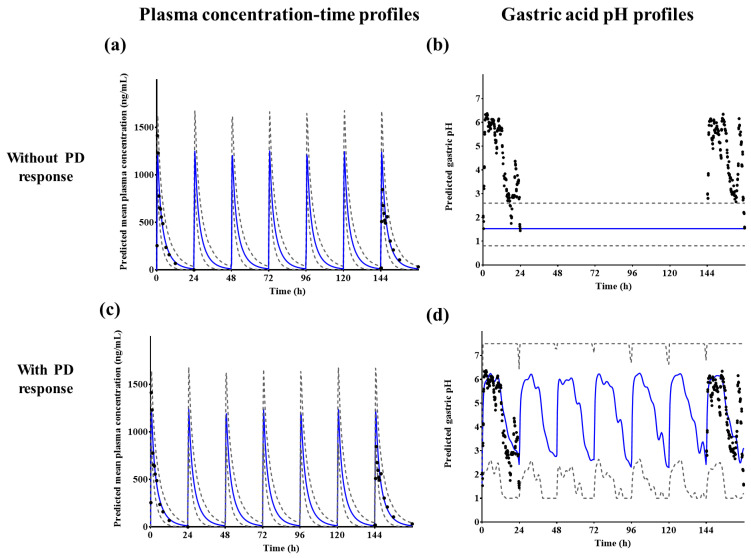
Predicted (**a**,**c**) plasma tegoprazan concentration and (**b**,**d**) gastric acid pH–time profiles after repeated administrations of 100 mg tegoprazan once daily for 7 days. Blue solid lines represent the predicted mean values, gray dotted lines represent the predicted 5th and 95th percentile, and black dots represent the observed plasma tegoprazan concentration or gastric acid pH profiles, respectively.

**Table 3 pharmaceutics-14-01298-t003:** Permeability and efflux ratio of tegoprazan from the Caco-2 permeability assay.

7.4/7.4 (pH)		Recovery (%)	Mean	Transport Rate (nmol/h)	P_app_ (×10^−6^ cm/s)	Mean	SD	Efflux Ratio
A to B	A	82.7	84.5	0.56	13.36	12.53	1.32	0.93
	B	86.4		0.48	11.01			
	C	84.5		0.56	13.23			
B to A	A	85.9	84.3	0.49	11.36	11.69	1.53	
	B	88.7		0.46	10.36			
	C	78.3		0.53	13.36			

A to B, apical to basolateral permeability; B to A, basolateral-to-apical efflux; P_app_, apparent permeability.

**Table 4 pharmaceutics-14-01298-t004:** The observed and predicted arithmetic mean AUC_last_, C_max_, CL values, and their ratios for tegoprazan and M1.

Parameters	Observed (n = 12)	Predicted (n = 1000)	Ratio (Predicted/Observed)
*Tegoprazan*			
AUC_0–24 h_ (ng × h/mL)	5611 (3783–7489)	5145 (2819–8135)	0.90 (0.85–0.96)
C_max_ (ng/mL)	1425 (908–2184)	1276 (896–1708)	0.88 (0.80–1.00)
CL (L/h)	17.8 (13.1–24.4)	21.5 (12.3–35.5)	1.25 (1.17–1.34)
T_max_ (h)	1.0 (0.5–1.0)	0.55 (0.3–1.6)	-
*M1*			
AUC_0–48 h_ (ng × h/mL)	8334 (6020–10,834)	7319 (4587–11,151)	0.88 (0.84–0.94)
C_max_ (ng/mL)	397.9 (238.9–575.5)	395.9 (142.1–1103.3)	1.01 (0.95–1.08)
T_max_ (h)	4.0 (3.0–8.0)	6.3 (1.2–17.6)	-

All parameters are presented as the arithmetic mean (5th–95th percentile), except T_max_, which is presented as the median (minimum–maximum value). The acceptance criteria for C_max_ and AUC_last_ ratio were within 30%. AUC: area under the curve from time 0 to the last sampling time point (for tegoprazan and M1, these AUCs were for 0–24 h and 0–48 h, respectively); C_max_, maximum concentration; CL, clearance; T_max_, time of maximum drug concentration

**Table 5 pharmaceutics-14-01298-t005:** The predicted and observed AUC_last_, C_max_, and T_max_ for tegoprazan and M1 after a single administration of 50 mg tegoprazan in the fasted state or at 30 min after a high-fat meal.

	Fast			At 30 min after a High-Fat Meal		
	Observed (n = 12)	Predicted (n = 1000)	Ratio (Predicted/Observed)	Observed (n = 12)	Predicted (n = 1000)	Ratio (Predicted/Observed)
*Tegoprazan*						
AUC_0–24 h_ (ng × h/mL)	2837 ± 1055	2533 ± 813.0	0.89 (0.45–1.33)	3017 ± 1194	2640 ± 828.8	0.88 (0.43–1.32)
C_max_ (ng/mL)	803.0 ± 156.2	684.9 ± 130.5	0.85 (0.62–1.09)	492.0 ± 293.9	470.7 ± 134.2	0.96 (0.32–1.59)
T_max_ (h)	1.0 (0.5–2.0)	0.6 (0.3–1.3)	-	3.0 (0.5–8.0)	1.05 (0.3–3.5)	-
*M1*						
AUC_0–48 h_ (ng × h/mL)	4462 ± 1008	3698 ± 989.4	0.83 (0.54–1.12)	3797 ± 915.6	3695 ± 1016	0.97 (0.62–1.33)
C_max_ (ng/mL)	198.8 ± 58.26	203.7 ± 154.6	1.02 (0.19–1.86)	142.2 ± 30.21	191.1 ± 134.4	1.34 (0.36–2.33)
T_max_ (h)	8.0 (3.8–8.0)	6.05 (1.1–17.1)	-	8.0 (6.0–24.0)	7.23 (1.8–20.2)	-

All parameters are presented as arithmetic mean ± standard deviation, except for T_max_, which is presented as the median (minimum–maximum value). The ratios of each parameter were calculated as follows: predicted mean/observed mean. The observed parameters were obtained from [34]. AUC, area under the concentration–time curve; C_max_, maximum concentration; T_max_, time of maximum drug concentration.

**Table 6 pharmaceutics-14-01298-t006:** The observed and predicted time of pH ≥ 4 over 24 h after single or repeated administrations of tegoprazan.

Parameters	50 mg Single		100 mg q.d. for 7 Days	
	Observed (n = 6)	Predicted (n = 1000)	Observed (n = 6)	Predicted (n = 1000)
pH ≥ 4 over 24 h (%)	48.9	49.6	Day 1: 62.3 Day 7: 70.4	Day 1: 63.0 Day 7: 63.0

Observed parameters were obtained from [15]. q.d.: once daily.

## Data Availability

Not applicable.

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
