# Peer review of "Integration of a Physiologically Based Pharmacokinetic and Pharmacodynamic Model for Tegoprazan and Its Metabolite: Application for Predicting Food Effect and Intragastric pH Alterations"

_pharmaceutics, 2022, doi:10.3390/pharmaceutics14061298_

Round 1

Reviewer 1 Report

Hyeon-Cheol Jeong et al integrated and applied the PBPK/PD model for Tegoprazan and its metabolite for predicting the food effect and intragastric pH alterations. The authors conducted permeability, and drug metabolism assays and used clinical PK and PD data to establish model development.

Below are the comments and include the description in the manuscript.

  1. In metabolic stability assay: which buffer and pH of the buffer were used for HLM metabolic stability studies? What is the percent of organic content in the reaction mixture?
  2. In CaCo2 permeability assay: what was the qualification criteria for CaCo2 cells before performing the permeability studies? What was the range of TEER values for CaCo2 cells observed? Line 170-171, how the cumulative amount of drug transported was calculated if you are using fresh HBSS for every time point?
  3. Why did the authors use carbamazepine as an internal standard in metabolic stability assays for tegoprazan and M1? Why didn’t you use d6-tegoprazan as an internal standard for these assays?
  4. The authors need to provide a description and results of bioanalytical method development and validation of tegoprazan using LCMS in the 2.5 section. LC-MS/MS acronym should be uniform throughout the manuscript.
  5. The authors should represent the figure with the structural compartments of the PBPK/PD model used for tegoprazan and its metabolite.
  6. How the PBPK models were developed for drug distribution, perfusion limited or permeability limited?
  7. How the model refinement, sensitivity analysis, and internal and external validation was performed?
  8. The equations mentioned in the results section should be incorporated in the materials and methods section.
  9. In figure 2 legend, delete “rate”. The number of replicates should be mentioned.
  10. In figure 3 legend, n=24 was mentioned for observed data but in table 4, n=12 for observed data. The authors conducted the clinical studies with 12 subjects in the 2.6 clinical PK dataset. Clarify this.
  11. IRB statement is applicable for this manuscript. The authors should include the IRB statement after the funding section.
  12. In Table S1, how many replicates were performed for this study? The authors can include the mean +/_ SD values in the table.

Author Response

  1. In metabolic stability assay: which buffer and pH of the buffer were used for HLM metabolic stability studies? What is the percent of organic content in the reaction mixture?

Answer: We used the buffer of 0.1 M potassium phosphate (pH 7.4). The percent of organic content was 0.01% of DMSO and less than 1% of acetonitrile in the reaction mixture. We have corrected the description and added a sentence in Materials and methods section.

Before: (Page 4, line 144-145) The NADPH-generating system, glucose 6-phosphate (3.3 mM), and human liver microsomes (HLM) (0.5 mg/mL) were combined and incubated at 37°C for 5 min.

After: (Page 4, line 144-147) The NADPH-generating system, glucose 6-phosphate (3.3 mM), 0.1 M potassium phosphate (pH 7.4), and human liver microsomes (HLM) (0.5 mg/mL) were combined and incubated at 37°C for 5 min. The organic content in the reaction mixture was 0.01% of DMSO and less than 1% of acetonitrile.

  1. In CaCo2 permeability assay: what was the qualification criteria for CaCo2 cells before performing the permeability studies? What was the range of TEER values for CaCo2 cells observed?

Answer: Before caco-2 assay, TEER value was evaluated to check the integrity of cell junctions. Acceptance criteria for the TEER values was higher than 300 Ω∙cm2. The changes of TEER values for caco-2 cells used in the experiment were within 10%. Description has been added as follows.

Before: (Page 4, line 166) Alterations were below 10% in all cases

After: (Page 4, line 172-175) Caco-2 cells with TEER values of higher than 300 Ω∙cm2 were used for permeability experiments and alterations of TEER value were below 10% in all cases (i.e., 417.3±6.3 Ω∙cm2 and 400.8±3.5 Ω∙cm2, before and after the permeability experiments, respectively).

Line 170-171, how the cumulative amount of drug transported was calculated if you are using fresh HBSS for every time point?

Answer: Cumulative transported amount was calculated through the sum of amount obtained from every time points. Description has been added as follows.

After: (Page 5, line 186-192) Cumulative transported amount was calculated as the sum of transported amounts for 0–15 min, 15–30 min, 30–45 min, and 45–60 min and the transport rate of tegoprazan was the slope of the regression line from the cumulative transported amount versus time plot. The apparent permeability (Papp) was calculated from the following equation (2).

  1. Why did the authors use carbamazepine as an internal standard in metabolic stability assays for tegoprazan and M1? Why didn’t you use d6-tegoprazan as an internal standard for these assays?

Answer: The metabolic stability assay (carbamazepine as an IS) and the quantification method validation and analysis (d6-tegoprazan as an IS) were performed in two different institutes. Although d6-tegoprazan was not used as an internal standard for metabolic stability assay, carbamazepine was successfully applied as IS in the assay.

  1. The authors need to provide a description and results of bioanalytical method development and validation of tegoprazan using LCMS in the 2.5 section. LC-MS/MS acronym should be uniform throughout the manuscript.

Answer: We have added the validation results of tegoprazan. The acronym for LC-MS/MS was unified throughout the manuscript.

Before: (Page 5, line 192-198) Caco-2 and rCYP samples were quantified using UPLC® I-class coupled with TQ-MS (Waters, Milford, MA, USA). The analytical method was validated based on the “Bioanalytical Method Validation Guidance for Industry” published by the US Food and Drug Administration and the “Guideline on Bioanalytical Method Validation” published by the Korean Ministry of Food and Drug Safety (MFDS) [32,33]. Linearity was in the range of 0.1–20 µM and the inter-batch accuracy and precision were in the range of 91.8%–101.3% and 2.3%–8.9%, respectively.

After: (Page 5-6, line 210-227) Caco-2 and rCYP samples were quantified using LC-MS/MS (UPLC® I-class coupled with TQ-MS, Waters, Milford, MA, USA). The mobile phase consisted of (a) 0.1% formic acid and (b) acetonitrile containing 0.1% formic acid. Gradient elution at a flow rate of 0.3 mL/min was applied. Tegoprazan, M1, and internal standard (tegoprazan-d6) were optimized in electrospray ionization positive mode. Acetonitrile with 0.1 μM of tegoprazan-d6 was added into collected caco-2 or rCYP samples. Then, processed samples were centrifuged at 13,000g for 20 minutes. After centrifugation, 100 μL of supernatant was transferred to a LC-MS vial and analyzed using LC-MS/MS. The analytical method was validated based on the “Bioanalytical Method Validation Guidance for Industry” published by the US Food and Drug Administration and the “Guideline on Bioanalytical Method Validation” published by the Korean Ministry of Food and Drug Safety (MFDS) [32,33]. Linearity was in the range of 0.1–20 µM and the inter-batch accuracy and precision were in the range of 91.8%–101.3% and 2.3%–8.9%, respectively. The stabilities of tegoprazan were evaluated and satisfied the acceptance criteria (coefficient of variances were within 15%) under variable conditions including reinjection, processed sample stability (PSS), short-term stability for stock and sample.

  1. The authors should represent the figure with the structural compartments of the PBPK/PD model used for tegoprazan and its metabolite.

Answer: Thank you for your suggestion. We have changed the figure 1 for structural compartments of the PBPK/PD model.

After: (Figure 1) The model structure of the tegoprazan-M1 PBPK/PD model

  1. How the PBPK models were developed for drug distribution, perfusion limited or permeability limited?

Answer: Perfusion-limited model was applied in the model. The perfusion rate limited model is applied for hydrophobic small molecule drugs, and the permeability rate limited model is used for low permeability or larger hydrophilic molecule drugs (Jones et al., 2013). The molecular weight of tegoprazan is 387.38 g/mol and the LogP of tegoprazan is 2.3. The Peff was 12.53 ±1.32× 10-6 cm/s in the Caco-2 permeability assay.

Reference: Jones, H. M. et al., CPT: pharmacometrics & systems pharmacology 2.8 (2013): 1-12.

  1. How the model refinement, sensitivity analysis, and internal and external validation was performed?

Answer: Model refinement was performed using a parameter estimation which was built into SimCYP® using observed tegoprazan and M1 concentration-time profiles after single administration of 100 mg tegoprazan. If the predicted concentration-time profiles and PK parameters were not fitted well to the observed profiles, the parameter estimation was employed. The appropriateness of the estimation results was evaluated through VPC and goodness-of-fit. Sensitivity analysis was performed to evaluate the initial values for parameter estimation. For internal validation step, when the arithmetic mean ratios of the predicted-to-observed PK parameters were within 30% range (0.7-1.3), we decided that the PBPK model was well fitted. After the final model was established, we performed external validation using observed PK/PD data after single administration of 50 mg tegoprazan and observed tegoprazan concentration-time profiles after repeated administrations of 100 mg tegoprazan.

  1. The equations mentioned in the results section should be incorporated in the materials and methods section.

Answer: Thank you for the suggestion. The equations were incorporated in the Methods section.

After:

(Page 4, line 156-160) The terminal slope of linear regression was used to calculate the intrinsic clearance (CLint) of tegoprazan and M1. For the in vitro HLM system, CLint was calculated using Equation (1).

CLint,mic (μL/min/mg protein) = k × Vincubation/Cincubation      (1)

where Vincubation is the incubation volume and Cincubation is the concentration of the microsomal protein.

(Page 5, line 192-194) The apparent permeability (Papp) was calculated from the following equation (2):

Papp (10-6 cm⁄s) = (transport rate (nmol/min)) / (concentration (μM) × area (cm2) × 60 s)             (2)

  1. In figure 2 legend, delete “rate”. The number of replicates should be mentioned.

Answer: Thank you for the comment. We have corrected the legend of figure 2.

Before: (Figure 2) The percent remaining rate of (a) tegoprazan and (b) M1 following incubation with human liver microsomes (HLMs) and the nicotinamide adenine dinucleotide phosphate (NADPH) cofactor.

After: (Figure 2) The percent remaining of (a) tegoprazan and (b) M1 following incubation with human liver microsomes (HLMs) and the nicotinamide adenine dinucleotide phosphate (NADPH) cofactor.

  1. In figure 3 legend, n=24 was mentioned for observed data but in table 4, n=12 for observed data. The authors conducted the clinical studies with 12 subjects in the 2.6 clinical PK dataset. Clarify this.

Answer: Thank you for pointing this out. We have checked and corrected.

Before: (Figure 3) Observed (black symbol, n = 24) and predicted (n = 1000, blue solid line) plasma (a) tegoprazan and (b) M1 concentration-time profiles after a single administration of 100 mg tegoprazan.

After: (Figure 3) Observed (black symbol, n = 12) and predicted (n = 1000, blue solid line) plasma (a) tegoprazan and (b) M1 concentration-time profiles after a single administration of 100 mg tegoprazan.

  1. IRB statement is applicable for this manuscript. The authors should include the IRB statement after the funding section.

Answer: We have added an IRB statement.

After: (Page 19, Institutional Review Board Statement). The clinical study protocols were reviewed and approved by the Institutional Review Board of Chonbuk National University Hospital (approval No.: CUH 2015-03-007).

  1. In Table S1, how many replicates were performed for this study? The authors can include the mean +/_ SD values in the table.

Answer: All experiments were performed in duplicate. To obtain the intrinsic clearance, the slope was calculated using the average of the peak areas of the duplicated samples. Thus the standard deviations could not be calculated for each intrinsic clearance of rCYPs. We have added a description.

Before: (Page 5, line 188-190) The concentrations of the remaining tegoprazan and M1 were determined by LC–MS/MS.

After: (Page 5, line 203-205) The concentrations of the remaining tegoprazan and M1 were determined by LC–MS/MS. All experiments were performed in duplicate.

Reviewer 2 Report

This is a nice paper describing the application of PBPK/PD modeling to describe tegoprazan and its metabolite pharmacokinetics under different conditions: (i) in pre- vs. postprandial states, (ii) after single and repeated drug administration, (iii) under the altered physiological conditions i.e., tegoprazan induced changes in gastric pH values. These kinds of in silico studies have proven to be highly useful in different stages of formulation development, and this may also apply for a PBPK/PD tegoprazan model.

The study is well designed and the results are generally clearly presented, although at some point the story is difficult to follow. The model prediction results fit the in vivo observed values fairly well; however, there are certain differences that should be elaborated in more detail. Also, a number of input model parameters have been optimized and the authors should justify these steps in the manuscript. Therefore, I do have a few comments and suggestions:

Major:

1. The major question concerns the selection of input parameters (Tables 1 and 2) where a number of parameters have been optimized (as the authors state “using experimental values and parameter estimation”). This includes permeability value, CLint for CYP 3A4 (which is more than twice higher than the experimental value) and CYP 2C19 for tegoprazan, and majority of inputs for M1. The authors should justify the optimization steps and explain the differences between the initial and optimized values. You should also comment that the employment of so many optimized input parameters (the need for massive parameter optimization) may be regarded as a limitation of the designed model.

In addition, I assume that the permeability value in Table 1 refers to human jejunal permeability? Please, clarify this in the table, and explain how did you convert the experimental Caco-2 permeability to human permeability value.  Also, you are stating that „(Peff,man), which fits well with the clinical study results” – How does this value fits with the clinical study results? Do you have experimental estimate for human Peff?

Also, please check (and explain and uniform) all abbreviations. E.g. you are mentioning Peff,man in Line 229 and Peff below Table 1 (Line 250), but this abbreviation is not mentioned in the Table). Also, you are mentioning Vd in the text (Lines 233 and 234), while Vss is stated in Table 1. You have stated „…adjusted by Vd in the clinical study, calculated from the NCA and parameter estimation…“ What was Vd value calculated by noncompartmental analysis? Please, provide the initial values of all model parameters in the manuscript.

2. Section 3.1: You need to provide comments on all the experimental results. E.g., you can comment on the major drug transport mechanism based on the obtained Papp (A-B) and (B-A) values and calculated efflux ratio.

3. Tables 4 and 5: Please, comment on the differences between the predicted and observed values. There are relatively large differences in Tmax values and you should elaborate on the possible reasons for these discrepancies.

4. Discussion section/Lines 501-515: You should clarify that the hypothesized changes in drug performance following repeated administration have not been reflected in the designed model.  

Minor:

5. A suggestion is to change the title of sections 2.8 to „Model verification“

6. Please, adjust the titles of sections 2.10 and 3.6 (a suggestion is„…gastric pH changes induces by tegoprazan“ or „…tegoprazan-induced gastric pH changes“)

7. Line 335: Please, explain the „Error!“ statement.

8. Lines 364-366: You are stating „The predicted mean plasma tegoprazan and M1 concentration-time profiles at a dose of 100 mg were fitted to the observed profiles (Figure 4).“ The observed values for M1 are not shown, so please correct.

9. Fig. 4: Indicate in the figure legend that dots represent the observed values for the parent drug.

10. Line 374: Please clarify the statement „The ratios of tegoprazan AUC0-24h and Cmax were within the range of 20%“ What do the ratios refer to (predicted vs. observed in the fasted state)? What does the range of 20% refer to i.e., 20% of what? The same applies for the similar statements in the text.

11. Lines 529-530: „When a high-fat diet or regular diet was used in the simulation…“ You have not shown the data that refer to the regular diet, so please correct.

12. Correct the numbers of tables mentioned in the Discussion section i.e., Line 531: Tmax and Cmax from clinical studies are not shown in Table 2; Lines 534 and 538: Table S2 instead of Table S3; Line 540: Table S3 instead of S4 (you only have 3 tables in the supplement)

13. Please clarify the statements in lines 542-546 (it’s not clear what the authors meant to say).

14. Conclusion/Lines 565-566: Please indicate to which GI region the „mean residence time“ refers to.

Author Response

Major:

  1. The major question concerns the selection of input parameters (Tables 1 and 2) where a number of parameters have been optimized (as the authors state “using experimental values and parameter estimation”). This includes permeability value, CLint for CYP 3A4 (which is more than twice higher than the experimental value) and CYP 2C19 for tegoprazan, and majority of inputs for M1. The authors should justify the optimization steps and explain the differences between the initial and optimized values. You should also comment that the employment of so many optimized input parameters (the need for massive parameter optimization) may be regarded as a limitation of the designed model.

Answer: For optimization steps, we optimized the permeability and intrinsic clearances for CYP3A4 and 2C19 using “parameter estimation” function in SimCYP®. Parameter estimation was performed using observed concentration-time profiles. The weighted least square was employed for objective function and the Nelder-Mead method was applied for minimization method of objective function value. Weighted least square parametric point was estimated using the following formula to determine the parameter value θ that minimizes the error value for each observation.

(ith subject who has nj observations; wij: the weights assigned at the jth observation)

The Nelder-Mead method is a method of local minimization that can find the minimum or maximum value of an objective function (Türkşen, Özlem et al., 2016).

Optimized caco-2 permeability and CYP3A4 and 2C19 CL were higher than the experimental values. The optimized caco-2 permeability was 7.98 times higher than experimental value, and the optimized intrinsic clearances of CYP3A4 and 2C19 were 2.25 times and 1.16 times higher than experimental values, respectively. In vitro-in vivo extrapolation was used to account for in vivo ADME processes in PBPK model (Chen, Yuan, et al., 2012). Multiple factors including solubility, dissolution rate, and gastrointestinal transit contribute to in vivo absorption (Lennernas, Hans., 2007). However, these factors are difficult to be determined precisely in in vitro assays. In a previously in vitro study, the caco-2 permeabilities of fexofenadine were increased approximately 2-3 times when fexofenadine was administered with p-gp inhibitors such as verapamil and ketoconazole. However, in vivo jejunal permeability in humans was not affected by p-gp inhibitors at clinical doses (Lennernas, Hans., 2007). Prediction of in vivo clearance using human liver microsomes was shown significant differences from actual values depending on the drugs (Li, Xue-Qing, et al., 2003).

Parameter estimation is widely used in studies of PBPK model development to overcome these limitations (eg. Tsamandouras et al., 2015; Pilla Reddy et al., 2018; Wang, Kun, et al., 2021; Umehara et al., 2019). Appropriate parameters were selected through statistical parameters, visual predictive check and goodness-of-fit. Parameter estimation is to predict and evaluate the parameters suitable for clinical study, not random values, using statistical techniques.

References:

Chen, Yuan, et al. Biopharmaceutics & drug disposition 33.2 (2012): 85-98.

SimCYP manual (unpublished)

Türkşen, Özlem, and Müjgan Tez. International Journal of Artificial Intelligence 14.1 (2016): 112-129.

Lennernas, Hans. Current drug metabolism 8.7 (2007): 645-657.

Li, Xue-Qing, et al. European journal of clinical pharmacology 59.5 (2003): 429-442.

Tsamandouras et al., British journal of clinical pharmacology 79.1 (2015): 48-55.

Pilla Reddy et al., CPT: pharmacometrics & systems pharmacology 7.5 (2018): 321-330.

Wang, Kun, et al., CPT: pharmacometrics & systems pharmacology 10.5 (2021): 441-454.

Umehara et al., Drug metabolism and personalized therapy 34.2 (2019)

In addition, I assume that the permeability value in Table 1 refers to human jejunal permeability?

Answer: In table 1, ‘Papp’ was apparent permeability. We have changed it to ‘Papp, caco-2’ for clear meaning.

In table 1, the intended meaning for Papp (apparent permeability) was caco-2 permeability. We have corrected the term clearly.

Before: (Table 1) Papp (×10−4 cm/s)

After: (Table 1) Papp,caco-2 (×10−4 cm/s)

Please, clarify this in the table, and explain how did you convert the experimental Caco-2 permeability to human permeability value. Also, you are stating that „(Peff,man), which fits well with the clinical study results” – How does this value fits with the clinical study results? Do you have experimental estimate for human Peff?

Answer: We have optimized the caco-2 permeability value using observed clinical study data. When caco-2 permeability was set to 10, predicted concentration-time profile was the most appropriate for fitting the clinical study data.

The built-in formula in SimCYP® for predicting Peff,man from caco-2 permeability (pH 7.4/7.4, passive & active) is as follows:

Also, please check (and explain and uniform) all abbreviations. E.g. you are mentioning Peff,man in Line 229 and Peff below Table 1 (Line 250), but this abbreviation is not mentioned in the Table).

Answer: We have checked all abbreviations throughout the manuscript and corrected the mentioned sentence clearly.

Before: (Page 5-6, line 227-230) The permeability of tegoprazan was obtained from the Caco-2 permeability assay and improved by estimating human jejunum permeability (Peff,man), which fits well with the clinical study results using parameter estimation.

After: (Page 6, line 257-259) The permeability of tegoprazan was obtained from the Caco-2 permeability assay and improved by parameter estimation, which fits well with the clinical study results.

Also, you are mentioning Vd in the text (Lines 233 and 234), while Vss is stated in Table 1. You have stated „…adjusted by Vd in the clinical study, calculated from the NCA and parameter estimation…“ What was Vd value calculated by noncompartmental analysis? Please, provide the initial values of all model parameters in the manuscript.

Answer: We have checked the abbreviations in manuscript. Observed Vd/F was obtained from previously reported clinical study literature (Hwang Jun Gi et al., 2019). SimCYP® calculates the volume of distribution at steady-state (Vss) (L/kg) using selected distribution model (full PBPK or minimal PBPK model). Initial value for Vss was 1.55 L/kg, however, predicted results shown well not fitted on observed profiles. We estimated Kp scalar in order to adjust the Vss using observed data and parameter estimation, estimated Kp scalar was applied in final model. Initial parameters and final parameters for tegoprazan and M1 models were presented in Table 1 and 2.

Reference:

Hwang, Jun Gi, et al. Translational and Clinical Pharmacology 27.2 (2019): 80-85.

  1. Section 3.1: You need to provide comments on all the experimental results. E.g., you can comment on the major drug transport mechanism based on the obtained Papp (A-B) and (B-A) values and calculated efflux ratio.

Answer: In the caco-2 permeability assay, the calculated recovery showed that any experimental error was not observed. Since the calculated efflux ratio was less than 2.0, the effect of p-gp might be insignificant. We have added sentences for explanations of caco-2 assay to Section 3.1.

After: (Page 10, line 341-344) The recovery calculated through the analysis results and the Papp value satisfied the range of ± 20%, therefore, we confirmed that it is no error during experiment processes. And, since efflux ratio was lower than 2, the effect of transporting by p-gp is expected to be insignificant.

  1. Tables 4 and 5: Please, comment on the differences between the predicted and observed values. There are relatively large differences in Tmax values, and you should elaborate on the possible reasons for these discrepancies.

Answer: Tmax could be influenced by physiological factors (i.e., alteration of gastric emptying time or gastric pH) or drug-specific factors (i.e., poor assumption for formulation or solubility). Tegoprazan is considered to be BCS class II drug because it has high permeability (experimental caco-2 permeability: 12.53 ± 1.32 × 10−6 cm/s) and low solubility (0.02 mg/mL in pH 6.8; source: patent). In the PBPK modeling study of BCS class II drug, colonic absorption, bile micelle solubilization and unbound fraction in gut enterocytes (fugut) have been identified as an important factor in considering absorption process (Sinha et al., 2012). In particular, fugut has had a significant impact on Cmax and AUC. In addition, PBPK modeling study for compound with pH-dependent solubility showed that a particle size could significantly influence on drug exposure (Chung et al., 2015).

In the current model, an ADAM absorption and diffusion layer model was applied to consider the formulation and solubility profiles. The diffusion layer model (DLM) was used to consider the properties of the formulation and hydrodynamic properties during drug absorption process. When the DLM was applied, bile solubility and formulation-related parameters such as particle size, radius, density, and supersaturation ratio were required. The default formulation parameters and predicted bile micelle mediated solubility were applied to the model due to the limited information. This limitation might affect to the differences of Tmax values. We have added the statement as follows.

After: (Page 17-18, line 545-559) Moreover, dissolution-related parameters such as supersaturation ratio, and precipitation rate constant might influence on decreases of Cmax values. Tegoprazan showed high permeability (experimental caco-2 permeability: 12.53 ± 1.32 × 10−6 cm/s) and low solubility (0.02 mg/mL in pH 6.8) and it was considered biopharmaceutics classification system (BCS) class II drug. In PBPK study for BCS class II drug, the colonic absorption, bile micelle solubilization and unbound fraction in gut enterocytes (fugut) have been identified as important factors for absorption process. Other PBPK study for a compound with pH-dependent solubility, a particle size was considered significant factor for drug exposure. Although DLM was applied in absorption module to apply the effect of formulation in the current model, the default values provided in SimCYP® for formulation-related parameters such as particle size, radius, density, and supersaturation ratio were applied due to lack of information. This might make some discrepancies in absorption process and related parameters.

References:

Composition for injection, https://patents.google.com/patent/KR20190005674A/en?oq=KR20190005674A

Sinha, Vikash Kumar, et al. Biopharmaceutics & drug disposition 33.2 (2012): 111-121.

Chung, John, et al. Journal of Pharmaceutical Sciences 104.4 (2015): 1522-1532.

Charkoftaki, Georgia, et al. Basic & clinical pharmacology & toxicology 106.3 (2010): 168-172.

  1. Discussion section/Lines 501-515: You should clarify that the hypothesized changes in drug performance following repeated administration have not been reflected in the designed model.

Answer: Repeated administrations of tegoprazan might lead to increase the intragastric pH and to decrease Cmax of tegoprazan. In a previous PBPK study of YH4808 (a P-CAB drug), the increase in intragastric pH caused the decrease in Cmax (Lee, Hyun A., et al., 2019). Another previous study assessed the effect of meal timing on the PK profiles of tegoprazan, a pH-dependent solubility and high permeable property were suggested (Yoon, Deok Y., et al, 2021). In the current study, a PD model with a feedback function was built and applied to predict the intragastric pH in virtual population. However, compared with the observed profiles, absence of decrease in Cmax was observed after repeated administrations. The limited absorption process in the current model could influence on the discrepancies. We have added the statement as follows.

After: (Page 17-18, line 545-559) Moreover, dissolution-related parameters such as supersaturation ratio, and precipitation rate constant might influence on decreases of Cmax values. Tegoprazan showed high permeability (experimental caco-2 permeability: 12.53 ± 1.32 × 10−6 cm/s) and low solubility (0.02 mg/mL in pH 6.8) and it was considered biopharmaceutics classification system (BCS) class II drug. In PBPK study for BCS class II drug, the colonic absorption, bile micelle solubilization and unbound fraction in gut enterocytes (fugut) have been identified as important factors for absorption process. Other PBPK study for a compound with pH-dependent solubility, a particle size was considered significant factor for drug exposure. Although DLM was applied in absorption module to apply the effect of formulation in the current model, the default values provided in SimCYP® for formulation-related parameters such as particle size, radius, density, and supersaturation ratio were applied due to lack of information. This might make some discrepancies in absorption process and related parameters.

References:

Lee, Hyun A., et al. European Journal of Pharmaceutical Sciences 130 (2019): 1-10.

Yoon, Deok Y., et al. Clinical and Translational Science 14.3 (2021): 934-941.

Minor:

  1. A suggestion is to change the title of sections 2.8 to „Model verification“

Answer: Thank you for your suggestion. We have changed title.

Before: (Page 7, Section 2.8) Model simulation and verification

After: (Page 8, Section 2.8) Model verification

  1. Please, adjust the titles of sections 2.10 and 3.6 (a suggestion is„…gastric pH changes induces by tegoprazan“ or „…tegoprazan-induced gastric pH changes“)

Answer: Thank you for your suggestion. We have changed the titles of 2.10 and 3.6

Before: (Page 8, 13) Incorporation of a PD model to predict gastric pH changes by tegoprazan

After: (Page 8, 13) Incorporation of a PD model to predict tegoprazan-induced gastric pH changes

  1. Line 335: Please, explain the „Error!“ statement.

Answer: Thank you for pointing out. We have checked and deleted error.

  1. Lines 364-366: You are stating „The predicted mean plasma tegoprazan and M1 concentration-time profiles at a dose of 100 mg were fitted to the observed profiles (Figure 4).“ The observed values for M1 are not shown, so please correct.

Answer: We have checked and corrected the sentences.

Before: (Page 12, line 363-365) The predicted mean plasma tegoprazan and M1 concentration-time profiles at a dose of 100 mg were fitted to the observed profiles (Figure 4).

After: (Page 12, line 391-393) The predicted mean plasma tegoprazan concentration-time profiles at a dose of 100 mg were fitted to the observed profiles (Figure 4).

  1. Fig. 4: Indicate in the figure legend that dots represent the observed values for the parent drug.

Answer: We have revised the figure legend.

After: (Figure 4) Predicted plasma (a) tegoprazan and (b) M1 concentration-time profiles after repeated administrations of 100 mg tegoprazan once daily for 7 days. Blue solid lines and black dotted lines represent the mean predicted plasma concentrations and the 95th and 5th percentiles, respectively. Black dots represent the observed tegoprazan concentration profiles (n=6). Only the predicted profiles are presented for M1.

  1. Line 374: Please clarify the statement „The ratios of tegoprazan AUC0-24h and Cmax were within the range of 20%“ What do the ratios refer to (predicted vs. observed in the fasted state)? What does the range of 20% refer to i.e., 20% of what? The same applies for the similar statements in the text.

Answer: Thank you for the comment. We have checked and corrected sentence.

Before: (Page 12, line 373-375) The ratios of tegoprazan AUC0-24h and Cmax were within the range of 20%, which is consistent with previously reported results (Table 5). For M1, the AUC0–48h ratio was within a range of 20%; however, the Cmax ratio exceeded 20%.

After: (Page 12, line 402-405) The ratios of tegoprazan AUC0-24h and Cmax were within the 30% range of mean ratio, which is consistent with previously reported results (Table 5). For M1, the AUC0–48h ratio was within 30% range of mean ratio; however, the Cmax ratio exceeded 30% range.

  1. Lines 529-530: „When a high-fat diet or regular diet was used in the simulation…“ You have not shown the data that refer to the regular diet, so please correct.

Answer: We have corrected the sentence.

Before: (Page 17, line 528-530) When a high-fat diet or regular diet was used in the simulation, Tmax was delayed and Cmax decreased, which is consistent with the results of the clinical study (Table 2).

After: (Page 18, line 572-574) When a high-fat diet was applied in the simulation, Tmax was delayed and Cmax decreased, which is consistent with the results of the clinical study (Table 5).

  1. Correct the numbers of tables mentioned in the Discussion section i.e., Line 531: Tmax and Cmax from clinical studies are not shown in Table 2; Lines 534 and 538: Table S2 instead of Table S3; Line 540: Table S3 instead of S4 (you only have 3 tables in the supplement)

Answer: Thank you for pointing this out. We have checked and corrected.

Before:

(Page 17, line 529-530) Tmax was delayed and Cmax decreased, which is consistent with the results of the clinical study (Table 2).

(Page 17, line 537-539) The predicted mean residence time in the stomach in the group of administered after high-fat meal increased by approximately 170.4% compared to fasting (Supplementary Table S4).

After:

(Page 18, line 573-574) Tmax was delayed and Cmax decreased, which is consistent with the results of the clinical study (Table 5).

(Page 18, line 574-577) The predicted absorbed fraction for administered at 30 min after a high-fat meal in the duodenum increased by approximately 72.7% and decreased in other segments compared with the group of administered at fasting (Supplementary Table S2).

  1. Please clarify the statements in lines 542-546 (it’s not clear what the authors meant to say).

Answer: We have revised the sentence.

Before: (Page 17, line 541-545) Based on these results, the increase in intragastric pH following the administration of tegoprazan and food intake may also be influenced by postprandial PK profiles; however, the effects of physiological factors, such as mean residence time and gastric emptying time, on the gastrointestinal tract, also are considered to be the major factors.

After: (Page 18, line 585-590) Increases of intragastric pH after tegoprazan administration or food intake is thought to be the major cause of decreases of Cmax of tegoprazan. However, physiological factors (i.e., gastric and small intestine mean gastric residence time) and formulation-related parameters (i.e., particle size, bile micelle mediated solubility parameters) also can affect on postprandial PK profiles.

  1. Conclusion/Lines 565-566: Please indicate to which GI region the „mean residence time“ refers to.

Answer: Thank you for pointing this out. We have checked and corrected.

Before: (Page 18, line 563-565) Although the absorbed fraction of tegoprazan increased in some GI tract segments in the postprandial state, it was expected that the delay in mean residence time would have a greater effect.

After: (Page 19, line 608-610) Although the absorbed fraction of tegoprazan increased in some GI tract segments in the postprandial state, it was expected that the delay in gastric and small intestine mean residence time would have a greater effect.

Round 2

Reviewer 2 Report

I am pleased to see that the authors have resolved most of the issues. Still, I believe that the manuscript requires some minor corrections:

1. Regarding the first question on the selection of input parameters and the need for their optimization, the authors correctly replied that “Parameter estimation is widely used in studies of PBPK model development”; however, it is also known that a number of the optimized parameters should be minimized (in contrast, in this study a large number of parameters have been optimized), and in case there’s a need for extensive parameter optimization, the model should be verified by comparison with another set of in vivo data (not the same one used for parameter optimization). I’m aware that the lack of additional in vivo data may limit such an approach, but in such case the authors need to state in the manuscript possible limitations of the designed model. And appropriate discussion should be included in the revised manuscript (not just in the reply to the reviewer).

2. Considering your answer/correction on drug permeability „After: (Page 6, line 257-259) The permeability of tegoprazan was obtained from the Caco-2 permeability assay and improved by parameter estimation, which fits well with the clinical study results“…the sentence you propose is still misleading (it sounds like the permeability fits the (permeability value) obtained in clinical studies); you need to state that „…the permeability was fitted so that the predicted concentration-time profile fits the clinical study results“.

3. I am still concerned about the question on volume of distribution and I believe this has not been properly explained in the manuscript. If you have used Rodgers & Rowland prediction method 2 to calculate Vss (and later parameter estimation to determine Kp and adjust Vss), why do you mention Vd from a clinical study? I assume you have estimated Kp using the observed plasma concentration data (not Vd; lines 261-262 in the revised manuscript) and parameter estimation. However, if you choose to mention Vd/F from a clinical study, you need to provide the exact value in the manuscript.

4. Please, clarify the statements about ratios of the predicted vs observed values (lines 402-405 in the revised manuscript). You cannot state „ratios of tegoprazan AUC0-24h and Cmax” (you surely don’t mean AUC/Cmax); you need to correctly specify “The ratios of predicted vs observed tegoprazan AUC0-24h and Cmax values were within...”, “For 404 M1, the predicted vs observed AUC0–48h ratio was...” etc.

5. You still have mistakes in the numbers of tables; please, carefully check and correct! Line 580 in the revised text: Table S2 instead Table S3 (surface solubility is shown in Table S2); Line 583 in the revised text: Table S3 instead Table S4 (again, you only have 3 tables in the supplement!)

Author Response

I am pleased to see that the authors have resolved most of the issues. Still, I believe that the manuscript requires some minor corrections:

  1. Regarding the first question on the selection of input parameters and the need for their optimization, the authors correctly replied that “Parameter estimation is widely used in studies of PBPK model development”; however, it is also known that a number of the optimized parameters should be minimized (in contrast, in this study a large number of parameters have been optimized), And appropriate discussion should be included in the revised manuscript (not just in the reply to the reviewer) and in case there’s a need for extensive parameter optimization, the model should be verified by comparison with another set of in vivo data (not the same one used for parameter optimization). I’m aware that the lack of additional in vivo data may limit such an approach, but in such case the authors need to state in the manuscript possible limitations of the designed model.

Answer: It is recommended that the optimized parameters be minimized. However, some clearance-related parameters (e.g., bile clearance, human liver cytosol clearance, and additional systemic clearance) for tegoprazan and M1 were estimated owing to the limited information. Single adjusting compartment-related parameters of the minimal PBPK model (e.g., volume of the single adjusting compartment, Vsac; first-order rate constant for distribution to the single adjustable compartment, kin; first-order rate constant for distribution from the single adjustable compartment, kout) comprise non-physiological parameters and can be obtained through parameter estimation (Cohen‐Rabbie, Sarit, et al., 2021; Garcia, Luna Prieto, et al., 2018). Therefore, we could obtain optimized permeability and enzyme kinetic parameters and estimated additional clearance parameters using parameter estimation.

Different data were used for parameter estimation and external validation. We used single administration data (100 mg tegoprazan) in the model development and parameter estimation step, and repeated administration (100 mg tegoprazan) and single administration (50 mg tegoprazan) data were used for external validation.

We have added a discussion of the parameter estimation to the Discussion section.

After: (Page 19, Line 764–773) The current model was built with extensive parameter optimization. When in vitro Caco-2 permeability and the intrinsic clearance of rCYPs were applied to the model, predicted concentration–time profiles were different from those of clinical study data. Caco-2 permeability and rCYP intrinsic clearance were estimated using parameter estimation, and the optimized value was fitted to the observed clinical study data. Although the final model was improved with extensive parameter optimization, it was validated using separate clinical study data that were not used for model development and model improvement. Therefore, the tegoprazan–M1 PBPK model developed in this study is considered a reasonable model.

References:

Cohen‐Rabbie, Sarit, et al. The Journal of Clinical Pharmacology 61.11 (2021): 1493-1504.

Garcia, Luna Prieto, et al. Drug Metabolism and Disposition 46.10 (2018): 1420-1433.

  1. Considering your answer/correction on drug permeability „After: (Page 6, line 257-259) The permeability of tegoprazan was obtained from the Caco-2 permeability assay and improved by parameter estimation, which fits well with the clinical study results“…the sentence you propose is still misleading (it sounds like the permeability fits the (permeability value) obtained in clinical studies); you need to state that „…the permeability was fitted so that the predicted concentration-time profile fits the clinical study results“.

Answer: Thank you for your comment. We have revised this sentence accordingly.

Before: (Page 6, Line 254-256) The permeability of tegoprazan was obtained from the caco-2 permeability assay and improved by parameter estimation, which fits well with the clinical study results.

After: (Page 6, Line 287–290) The permeability of tegoprazan was determined by a Caco-2 permeability assay and optimized by parameter estimation. Caco-2 permeability was optimized such that the predicted concentration–time profiles were fitted to clinical study data.

  1. I am still concerned about the question on volume of distribution and I believe this has not been properly explained in the manuscript. If you have used Rodgers & Rowland prediction method 2 to calculate Vss (and later parameter estimation to determine Kp and adjust Vss), why do you mention Vd from a clinical study? I assume you have estimated Kp using the observed plasma concentration data (not Vd; lines 261-262 in the revised manuscript) and parameter estimation. However, if you choose to mention Vd/F from a clinical study, you need to provide the exact value in the manuscript.

Answer: The reason we explained the Vd of the clinical study is because we used this value in the initial model of tegoprazan. In addition, the observed Vd was used as an initial value to estimate the appropriate Kp scalar for the observed data. We have clearly revised the sentence to avoid confusion.

Before: (Page 6, Line 258-263) The distribution of tegoprazan was predicted using a full PBPK model. In this model, volume of distribution (Vd) varied according to the tissue:plasma partition coefficient (Kp) scalar values; therefore, the global Kp scalar was adjusted by Vd in the clinical study, calculated from the NCA and parameter estimation. The Rodger and Rowland model (method 2) was applied to consider the weak base property and distribution of the neutral and ionic forms.

After: (Page 6, Line 292–298) The tissue distribution in the PBPK model was predicted by the distribution model (e.g. Rodger and Rowland method, Poulin and Theil method) and the tissue:plasma partition coefficient (Kp) scalar for specific organ or global values. Predicted systemic distribution was expressed as the volume of distribution in a steady state (Vss). The Vd values of the clinical study were applied to the initial model of tegoprazan and parameter estimation was used to determine the optimized Kp scalar fitted to the observed data.

  1. Please, clarify the statements about ratios of the predicted vs observed values (lines 402-405 in the revised manuscript). You cannot state „ratios of tegoprazan AUC0-24h and Cmax” (you surely don’t mean AUC/Cmax); you need to correctly specify “The ratios of predicted vs observed tegoprazan AUC0-24h and Cmax values were within...”, “For 404 M1, the predicted vs observed AUC0–48h ratio was...” etc.

Answer: Thank you for pointing this out. We have checked this expression throughout the manuscript and corrected it for clarity.

Before:

(Page 8-9, Line 294-296) If the arithmetic mean ratio and its 95% CI were within the 30% range (0.7 – 1.3), the model was considered to fit well.

(Page 11, Line 369-371) The arithmetic mean ratios of AUClast, Cmax, and CL and the 95% confidence interval values were included in the 30% range of mean ratio.

(Page 13, Line 400-402) The ratios of tegoprazan AUC0-24h and Cmax were within the 30% range of mean ratio, which is consistent with previously reported results (Table 5). For M1, the AUC0–48h ratio was within 30% range of mean ratio; however, the Cmax ratio exceeded 30% range.

(Page 13, Line 402-403) however, the Cmax ratio exceeded 30% range.

(Page 15, Line 452-454) The AUC0–24h, Cmax, and CL arithmetic mean ratios (95% confidence intervals) for tegoprazan were 0.90 (0.85–0.96), 0.88 (0.80–1.00), and 1.25 (1.17–1.34), respectively.

After:

(Page 9, Line 345–347) If the predicted vs observed ratio and its 95% CI were within the 30% range (0.7–1.3), the model was considered to fit well.

(Page 11, Line 422–423) The ratios of predicted vs observed AUClast, Cmax, and CL and 95% confidence interval values were included in the 30% range of the mean ratio.

(Page 13, Line 464–466) The ratios of predicted vs observed tegoprazan AUC0-24h and Cmax values were within the 30% range of the mean ratio, which is consistent with previously reported results (Table 5). For M1, the ratio of the predicted vs observed AUC0–48h ratio was within the 30% range of the mean ratio

(Page 13, Line 467–468) however, the predicted vs observed Cmax ratio exceeded the 30% range.

(Page 16, Line 527–529) The ratios of predicted vs observed AUC0–24h, Cmax, and CL (95% confidence intervals) values for tegoprazan were 0.90 (0.85–0.96), 0.88 (0.80–1.00), and 1.25 (1.17–1.34), respectively.

  1. You still have mistakes in the numbers of tables; please, carefully check and correct! Line 580 in the revised text: Table S2 instead Table S3 (surface solubility is shown in Table S2); Line 583 in the revised text: Table S3 instead Table S4 (again, you only have 3 tables in the supplement!)

Answer: Thank you for pointing this out. We have checked the references to tables and figures throughout the manuscript and corrected them all accordingly.

Before:

(Page 18, Line 578-579) there was a slight decrease or no difference in the solubility in the remaining segments in the group compared to the fasted state (Supplementary Table S3).

(Page 18, Line 579-582) The predicted mean residence time in the stomach in the group of administered after high-fat meal increased by approximately 170.4% compared to fasting (Supplementary Table S4).

After:

(Page 18, Line 652–654) there was a slight decrease or no difference in solubility in the remaining segments in this group compared to that in a fasted state (Supplementary Table S2).

(Page 18, Line 654–657) The predicted mean residence time in the stomach upon administration of tegoprazan at 30 min after a high-fat meal was increased by approximately 170.4% compared to that administered while fasting (Supplementary Table S3).
